# Extraction of Copper and Zinc from Waste Printed Circuit Boards

**Ayorinde Emmanuel Ajiboye [1,2,\*], Folorunsho Emmanuel Olasehinde [1], Ojo Albert Adebayo [1], Olubode John Ajayi [1], Malay Kumar Ghosh [2] and Suddhasatwa Basu [2]**

[1]   Department of Chemistry Federal University of Technology, 340252-Akure, Nigeria
[2]   CSIR-Institute of Minerals and Materials Technology, Bhubaneswar 751013, India
[\*]   Correspondence: emmanuelayorinde2010@yahoo.com; Tel.: +23-4803-7304-122; Fax: +33-1427-321-20

**Abstract:** The recovery of valuable metals from waste printed circuit boards (WPCBs) is crucial in order to harness their economic resources, and prevents potential environmental contamination. However, selective extraction of Cu and Zn, and the co-extraction of other metals as impurities at ambient temperature using selected lixiviants such as HCl, $H_2SO_4$, $HNO_3$, trifluoromethanesulfonic acid (TFMS), NaOH, and mixtures of NaCl and $CuCl_2$ was studied. It is shown that the extraction efficiencies of all the metals increased with increases in lixiviant concentrations. High selectivity of Cu and Zn toward Fe were achieved in dilute $H_2SO_4$, $HNO_3$, TFMS, and 0.5 M NaCl + 0.1 M $CuCl_2$, and low dissolution of Pb (<5%) was observed in all $H_2SO_4$ lixiviants. Almost 100% Zn extraction using NaOH lixiviants without trace of other metals was achieved. Therefore, 0.5 M NaCl + 0.5 M $CuCl_2$, 1.0 M $HNO_3$, 0.5 M $H_2SO_4$, and 1.0 M TFMS showed high extraction selectivity toward Cu and Zn with low chemical consumption, and produced pregnant leach solution rich in Cu and Zn, as well as residue containing Fe, Ni, and other metals.

**Keywords:** extraction; copper; zinc; ambient temperature; lixiviants

## 1. Introduction

Presently, one type of waste with great environmental concern and economic value is electronic waste [1]. Although most metals are base metals such as copper, iron, nickel, tin, lead, aluminum, and zinc, a significant amount of attention has been expended on the recovery of precious metals, including gold, silver, and palladium [2]. Copper is widely used as a major interconnecting material in electronic industries, owing to its good electrical and thermal conductivity [3].

In recent years, several techniques have been employed for recycling activities, such as pyro-metallurgy [4], hydrometallurgy [5], bio-hydrometallurgy [6], etc. Various hydrometallurgical methods for the recovery of metals from printed circuit boards (PCBs) have been developed using leaching reagents such as acids, bases, and salts. Meanwhile, hydrometallurgical techniques [7], leaching [8], kinetic studies [9], and bioleaching of precious metals from waste electrical and electronic equipment (WEEE) using bacteria [6,10,11] have been attempted and reported in the literature.

On average, typical PCBs contain about 7.0% Fe, 27.0% Cu, 2.0% Al, 1.5% Zn, 0.5% Ni, 2000 ppm Ag, and 80 ppm of Au [12]. The leaching of metals from the PCBs by acids produces pregnant leach solutions containing several metals, including iron, which have been identified as the major problem in the electro-winning or electrochemical deposition of metals. For instance, in the electro-winning of Cu, it has been reported that Fe concentrations >30 ppm reduce current efficiency and increase the energy consumption in addition to changing the morphology of the cathodic Cu product due to Fe ions being oxidized from ferrous ($Fe^{2+}$) to ferric ($Fe^{3+}$) at the anode, and conversely reduce from $Fe^{3+}$ to $Fe^{2+}$ at the cathode, using up the required current and energy for the deposition of copper [13,14].

However, in order to avoid iron leaching into solution, the selective recovery of Cu using ammoniacal solutions and salts remains the most common and effective established method in hydrometallurgy [15]. The extraction of metals from ammonia solutions seems to be more complicated as compared to acidic environments [16,17]. The alkalinity of the solution and the free ammonia concentration affect the route of extraction [17]. However, copper-pregnant leached solutions require a further preceding process (solvent extraction) due to the complexation of Cu with ammonia, and require a solvent to push out the copper (Equations (1) and (2)), as this process has been reported to be ineffective [16,18].

$$M^{2+} + 4NH_3 \rightarrow M\,(NH_3)_4{}^{2+} \tag{1}$$

$$Cu\,(NH_3)_4{}^{2+}{}_{(aq)} + 2HA\,(o) \rightarrow CuA_2(o) + 2NH_4{}^{+}{}_{(aq)} + 2NH_{3(aq)} \tag{2}$$

With proper understanding of the principle of economic, eco-friendly, and efficiency (EEE) in the recycling of valuable metals from waste materials, the application of many further unit operations will make the recovery cumbersome and costly. Therefore, this study was carried out to investigate the selective extraction of Cu and Zn from waste printed circuit boards (WPCBs) using some selected lixiviants at ambient temperatures to produce pregnant leach solutions rich in Cu and Zn and residues rich in Fe and Ni, with other metals for further recovery. The lixiviants investigated for this purpose were: hydrochloric acid (0.5–5.0 M HCl), sulfuric acid (0.5–4.0 M $H_2SO_4$), nitric acid (1.0–5.0 M $HNO_3$), trifluoromethanesulfonic acid (1.0–2.5 M TFMS), sodium hydroxide (1.0–5.0 M NaOH), and mixtures of sodium chloride and copper (II) chloride salts (0.5 M NaCl + 0.1 M $CuCl_2$, 0.5 M NaCl + 0.5 M $CuCl_2$, 4.5 M NaCl + 0.1 M $CuCl_2$, and 4.5 M NaCl + 1.0 M $CuCl_2$).

## 2. Materials and Methods

### 2.1. Sample Preparation

Discarded WEEE for this research work was collected from waste store at the CSIR-Institute of Minerals and Materials Technology, Bhubaneswar, India, and unwanted attachments such as cords, wires, and plastic coverings were detached manually. Attached electronic components (ECs) on PCBs were removed by de-soldering using a hot air gun (Black & Decker KX 1800, Beijing, China). The pulverized sample obtained was screened using a sieve for particle size distribution, while the coarse sample was returned into the mill to obtain powder samples ≤100 μm.

Material Pretreatment by Roasting

For each batch of roasting, 100 g of pulverized WPCBs were roasted in a muffle furnace (Nabertherm Furnace, Lilienthal, Germany) at 700 °C for 2 h without any addition of roasting agent to convert metals into their oxide form and remove organic materials. The gas passed through aluminum oxynitride tube connected at the top corner of the furnace into an alkaline solution/distilled water chamber tank, where it was absorbed. Then, the roasted sample obtained was powdered using a mortar and pestle, sieved with 75-μm mesh, and stored in an airtight container.

### 2.2. Characterization

The particle size distribution (PSD) of the sample (Figure 1) was obtained by a Mastersizer 2000 laser diffraction particle size analyzer with a Scirocco 2000 Dry Powder Feeder, both manufactured by Malvern Instruments (Malvern, UK). The sample was characterized for elemental phases present (Figure 2) using X-ray diffraction (X'Pert Pro-PAN Analytical X-Ray Diffractometer (Model: EMPYREAN)) operating at an anode current of 40 mA at 45 kV with Cu K$\alpha$ radiation, by Rietveld refinement method using High Score Plus software. SEM-EDX analysis for the morphological and phase identifications (Figure 3) was performed with an EVO 18 scanning electron microscope (Carl Zeisis, Munich, Germany) and E1220 EDAX (Element Company, Pleasanton, CA, USA) with Team software. Tungsten filament was used

as a cathode, the acceleration voltage (EHT voltage) was 20 kV, and the working distance (WD) was 8.5 mm.

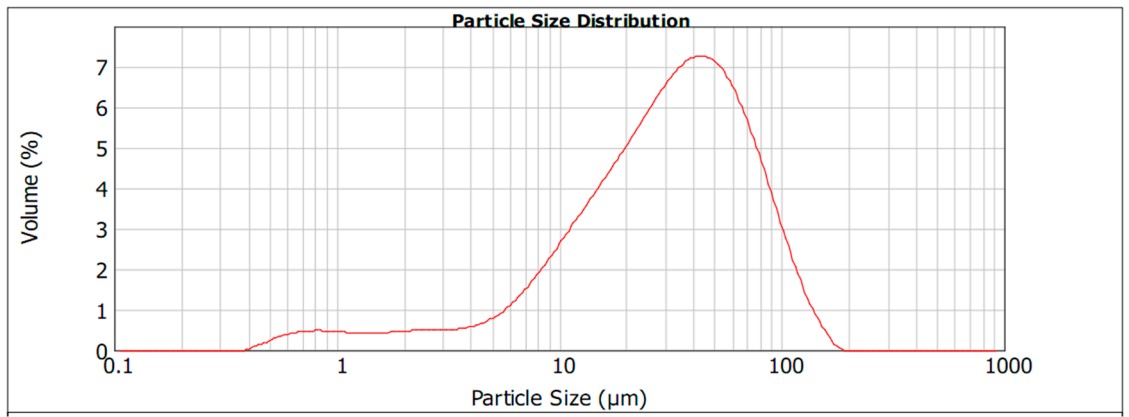

**Figure 1.** The particle size distribution of the powdered printed circuit board (PCB) sample.

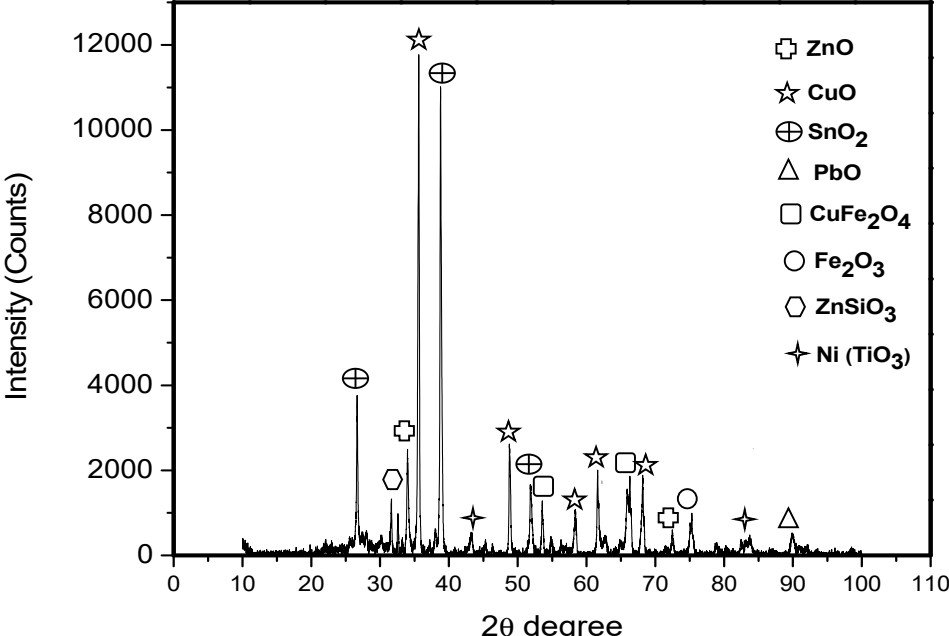

**Figure 2.** The observed X-ray diffraction (XRD) pattern of powdered PCBs sample.

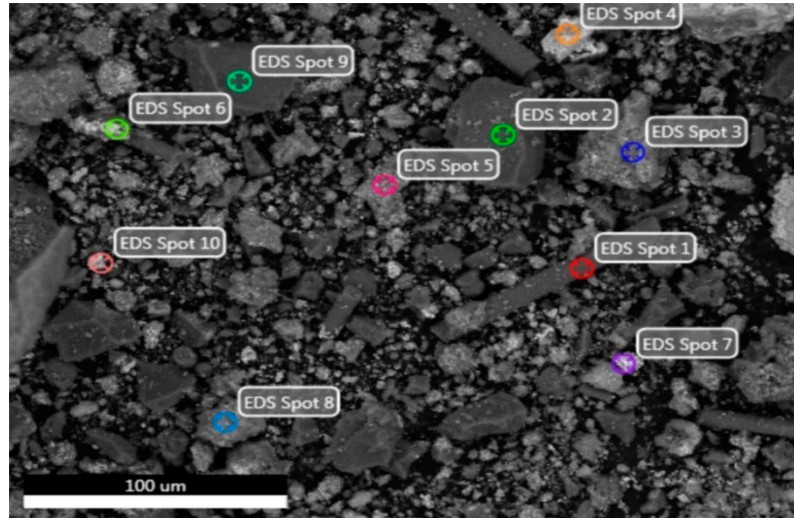

| Element | Average spot weight (%) |
|---------|-------------------------|
| **O K** | 34.93 |
| **AlK** | 4.57 |
| **SiK** | 16.24 |
| **PbM** | 0.67 |
| **SnL** | 3.18 |
| **CaK** | 9.13 |
| **TiK** | 0.93 |
| **CrK** | 0.33 |
| **FeK** | 3.95 |
| **YbL** | 0.92 |
| **NiK** | 0.11 |
| **CuK** | 20.93 |
| **OsL** | 1.62 |
| **PtL** | 1 |
| **AuL** | 1.5 |

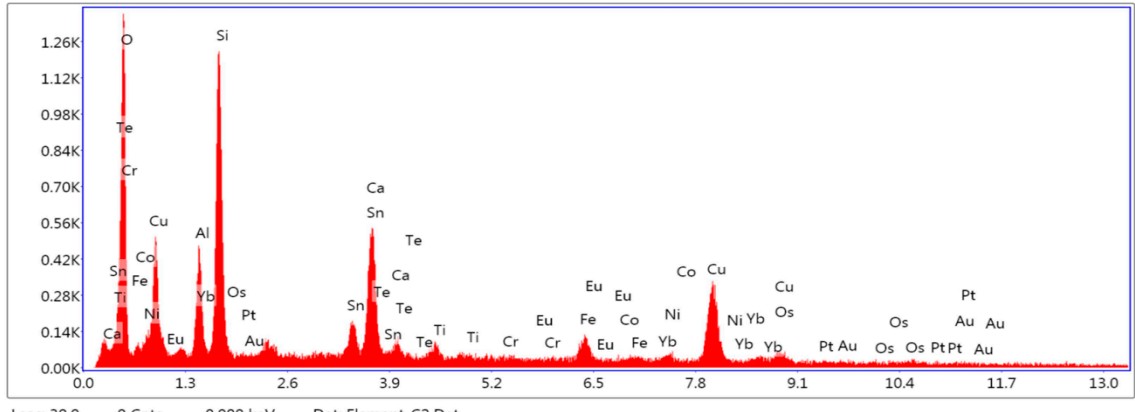

**Figure 3.** The scanning electron microscopy (SEM-EDX) of PCBs powder sample.

Chemical Analysis

Chemical analysis of WPCBs powder was performed after digestion using aqua regia. The concentration of tin (Sn) in the solution was analyzed by inductively coupled plasma optical emission spectroscopy (ICP-OES, Thermo Scientific, iCAP 7600 Duo SERIES, Waltham, MA, USA), while the concentration of other metals was analyzed using Atomic Absorption Spectrophotometer (AAS, AA200, Perkin Elmer, Shelton, CT, USA) (Table 1).

**Table 1.** Composition of metals in the waste printed circuit boards (WPCBs) powder sample (−75 μm).

| | Elements | Weight (%) |
|---|----------|------------|
| | Fe | 11.118 |
| | Cu | 41.64 |
| | Ni | 2.28 |
| | Al | 5.05 |
| | Zn | 1.79 |
| | Pb | 1.09 |
| | Sn | 1.63 |
| | Ti | 0.18 |

### 2.3. Experimental Studies

All chemicals and reagents used were of analytical grade, with the specifications given in Table 2. The extraction of metals from powder WPCBs by the lixiviants was conducted at ambient temperature (25 °C) for 12 h using a 0.5 L Erlenmeyer glass reactor connected to a thermostatic water bath without any external oxidation or the addition of an oxidizing agent. Mixing of the solutions was achieved using an MS-H280-pro magnetic stirrer (SCILOGEX, Rocky Hill, CT, USA) at 500 rpm.

The solid–liquid ratio employed was 0.05 (pulp density: 50 g/L). In order to evaluate the extraction efficiencies of Cu, Zn, and other metals (Equation (1)), the resulting liquors were filtered after leaching and the filtrate was analyzed by inductively couple plasma-optical emission spectroscopy (ICP-OES) and atomic absorption spectrophotometry.

$$\text{Extraction efficiency } (\%) = (Cm \times 10000)/(P \times Xm), \tag{3}$$

where $Cm$ is the concentration of metals in the leached solution in grams per liter (g/L); $P$ is the pulp density of the sample in percentage (%; i.e., g/100 mL); and $Xm$ is the metal content in the original sample (%; 1%= 10,000 g/L).

## 3. Results and Discussion

**Table 2.** Detailed information of the lixiviants used for the extraction studies.

| Solutions | Concentrations | Chemical Strength | Manufacturer |
|---|---|---|---|
| Hydrochloric Acid (HCl) | 0.5 M<br>1.5 M<br>2.5 M<br>4.5 M<br>5.0 M | Percent purity: 35.4%, specific gravity: 1.18 g/cm$^3$, percent weight: 36.46 g/mole | Finar Chemical |
| Sulfuric Acid (H$_2$SO$_4$) | 0.5 M<br>1.5 M<br>2.5 M<br>4.0 M | Percent purity: 98.0%, specific gravity: 1.84 g/cm$^3$, percent weight: 98.08 g/mole | Merck Specialities Private |
| Nitric Acid (HNO$_3$) | 1.0 M<br>2.5 M<br>3.5 M<br>5.0 M | Percent purity: 69.0%, specific gravity: 1.42g/cm$^3$, percent weight: 63.01 g/mole | HiMedia Laboratories Pvt. Ltd. |
| Trifluoromethanesulfonic Acid (TFMS) | 1.0 M<br>2.5 M | | |
| Sodium Hydroxide | 1.0 M<br>2.0 M<br>5.0 M | NaOH | |
| Copper (II) Chloride, pH 1 | 0.5 M NaCl + 0.1M CuCl$_2$<br>4.5M NaCl + 0.5 M CuCl$_2$<br>4.5 M NaCl + 0.1 M CuCl$_2$<br>4.5 M NaCl + 1.0 M CuCl$_2$ | CuCl$_2$ 2H$_2$O | |

### 3.1. Physical and Chemical Characterization of Pulverized WPCBs

According to the mineral analysis results of the powdered WPCBs sample (Figure 1), the main mineral phases in the sample of particle size $D_{50}$ (75 μm) were: CuO, Ni (TiO$_3$), NiCuO, ZnO, Zn$_2$SiO$_4$, SnO, CuFe$_2$O$_3$, and FeMn$_2$O$_4$. It is noteworthy that nickel formed a stable phase with TiO$_3$ and CuO, while iron was in oxide form. Additionally, zinc silicate and oxide were presents in the sample. It is expected that higher extraction of Cu and Zn will be achieved due to the fact that most oxides of metal are readily dissolve in dilute acid. Lower extraction of Ni is expected because of its stable metallic phase except in a concentrated acidic environment. In Table 1, elemental analyses revealed that the sample mainly contained Cu (41.64%), Zn (1.79%), Ni (1.8%), Fe (11.12%), Pb (2.09%), Al (5.05%),

and Sn (1.63%). These results reveal that Cu and Fe were the major metals in the WPCBs. Therefore, the degree of selective extraction of Cu and Zn with the co-extraction of Fe, Ni, Al Pb, and Sn using selected lixiviants at ambient temperature will be discussed in detail in the following paragraphs.

### 3.2. Extraction in Hydrochloric Acid

We studied the extraction of Cu and Zn, and the co-extraction of other metals from the pulverized WPCBs powder using varying concentrations of HCl lixiviant at ambient temperature for 12 h. However, the results presented in Figure 4 show that the extraction efficiency of Cu increased rapidly with increases in the concentration of HCl from 0.5 to 2.5 M. That is, the efficiency of Cu increased from 65.1% to 80.1% when the HCl concentration increased from 0.5 to 2.5 M. Further increases in the concentration from 2.5 to 5.0 M HCl had a negligible effect, as the efficiencies somewhat decreased because of incomplete dissociation equilibrium due to insufficient hydration.

On the other hand, an increase in the concentration of HCl was observed to have an insignificant effect on the extraction efficiencies of Zn, with 58.5% to 62.3% extracted when the HCl concentration was increased from 0.5 to 5.0 M, respectively. Except for Ni, the co-extraction of other metals (Fe, Ni, Al, Pb, and Sn) was observed with extraction efficiencies increasing proportionately with increases in HCl concentrations. The exception of Ni is due to stable phase with titanium oxides as $Ni(TiO_3)$, which is known for its high corrosion resistance [19]. It has been reported that higher concentration of HCl or temperature is required to achieve higher Sn extraction [20]. However, high extraction efficiency of Sn was observed when the concentration of HCl was increased from 2.5 to 4.5 M. Our results are in tandem with the report of [21], which indicated that the rate of dissolution of Sn increases with increasing hydrogen concentration. A further increase in the concentration to 5.0 M showed slight decrease in the extraction efficiency. This was as a result of formation of black surface film due to increasing hydrogen ion concentration at higher HCl concentration [21].

Therefore, it is important to note that the extraction of powdered PCBs sample by HCl lixiviant is not suitable for generating pregnant leach solution (PLS) that is rich in Cu and Zn since higher dissolution of other metals as impurities were observed.

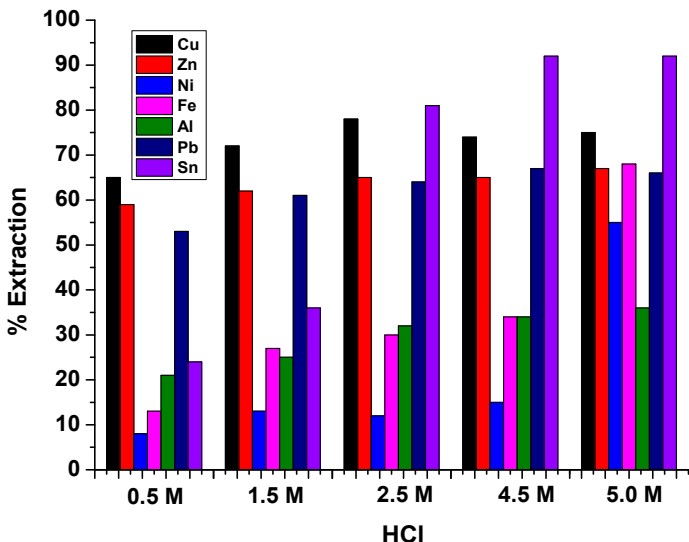

**Figure 4.** Comparative extraction efficiencies of Cu and Zn, and the co-extraction of other metals from WPCBs using HCl lixiviant, 50 g/L pulp density, 25 °C, 300 rpm, and 12 h.

### 3.3. Extraction in Sulfuric Acid

The degree of extraction of Cu and Zn, and the co-extraction of other metals from pulverized powdered WPCBs using different concentrations of $H_2SO_4$ lixiviant at ambient temperature for 12 h was also studied. In contrast to the results of HCl lixiviants, a high extraction efficiency of Zn was

achieved in the H$_2$SO$_4$ lixiviant investigated (Figure 5). This behavior was as result of the high corrosion potentials of H$_2$SO$_4$ (−0.98 mV.SCE) as compared to HCl (−1.00 mV.SCE) [22]. Yoshida reported similar results [23]. Further slight increase in the extraction efficiencies of Zn was observed when the concentration of H$_2$SO$_4$ lixiviant increased to 4.0 M. The low extraction of Pb was due to non-solubility in sulfuric acid, as reported in [24]. Proportionate increases in the extraction efficiency of iron with respect to increases in concentration of H$_2$SO$_4$ lixiviant were observed. In comparison to results of HCl lixiviants, the extraction efficiencies of iron and Sn were considerably low. This was as a result of the lower pKa of H$_2$SO$_4$ as compared to HCl. Thus, the pKa for HCl and H$_2$SO$_4$ were found to be −6.3 and −3.0, respectively. This implies that HCl dissociates faster than H$_2$SO$_4$ [25]. Additionally, the lower extraction of Fe in all the H$_2$SO$_4$ treatments was because the main phase of Fe was copper-magnetite (CuFe$_2$O$_3$), as it has been reported that temperature has a greater effect on the dissolution of magnetite with sulfuric acid than the acid concentration [26]. Generally, the dissolution of powdered WPCBs in 0.5 M H$_2$SO$_4$ would produce PLS rich in Cu and Zn, with low concentrations of Fe and other metals.

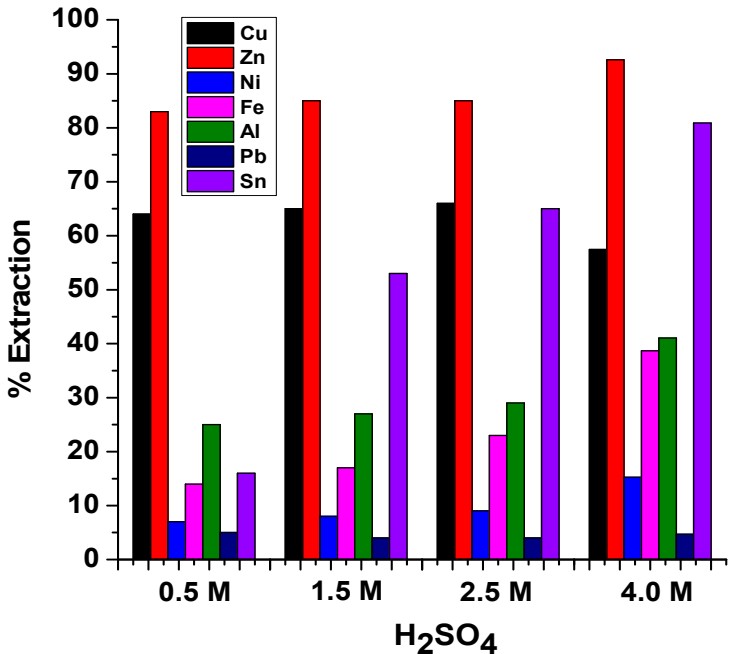

**Figure 5.** Comparative extraction efficiencies of Cu and Zn, and the co-extraction of other metals from WPCBs using H$_2$SO$_4$ lixiviant, 50 g/L pulp density, 25 °C, 300 rpm, and 12 h.

*3.4. Extraction Study in Nitric Acid*

Although nitric acid is well-known for its strong oxidizing power [27], we studied its efficiency for the extraction of Cu and Zn, and the co-extraction of other metals from powdered WPCBs at ambient temperature (Figure 6). It was observed that the extraction efficiency of Cu increased slowly with respect to increasing concentration from 1.0 to 3.5 M. A further increase in the concentration to 4.0 M did not have much effect on the extraction efficiency. Conversely, a high extraction efficiency of Zn was also achieved in HNO$_3$ lixiviants, except 4.0 M HNO$_3$, where the extraction of Zn was found to have decreased. The decreased in the dissolution efficiency might be a result of the poor dissociation of HNO$_3$ at high concentration.

The co-extraction of other metals was also explored. It was found that their extraction efficiencies increased with increasing in the concentrations of HNO$_3$ lixiviant. The extraction efficiencies of Ni and Fe increased from 5.1% and 6.7% to 40% and 15.8%, respectively, when the concentration of HNO$_3$ lixiviant increased from 0.5 to 4.0 M. The high extraction efficiency of Ni was due to the oxidizing power of HNO$_3$. However, 1.0 and 2.5 M HNO$_3$ lixiviants were observed to yield PLSs with 60% and

70% Cu and Zn, respectively. Additionally, the co-extraction of Fe and Ni in the PLS was observed to be <7.0%.

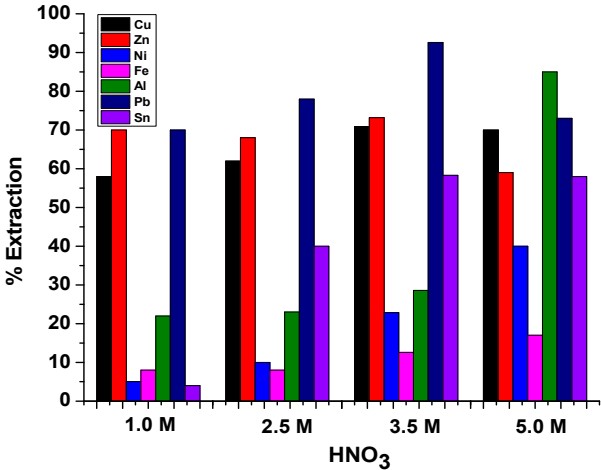

**Figure 6.** Comparative extraction efficiencies of Cu and Zn, and the co-extraction of other metals from WPCBs using HNO$_3$ lixiviant, 50 g/L pulp density, 25 °C, 300 rpm, and 12 h.

### 3.5. Extraction in Trifluoromethanesulfonic Acid (TFMS)

Organic acids (e.g., some sulfonic acids) are a class of strong acids with readily (bio)degradable conjugate bases that have been demonstrated as generally less environmentally persistent than mineral acids such as sulfuric acid [28].

Methanesulfonic acid, CH$_3$SO$_3$H (pKa = −1.9), has been demonstrated as being highly efficient for the dissolution of a number of different heavy metals via the formation of soluble methanesulfonate complexes [28]. Yet, TFMS with a pKa of −15 is also a derivative and has not been employed for the leaching of metals. However, a high extraction of Cu and Zn was achieved, although the extraction efficiencies were almost the same when the concentration of TFMS increased from 1.0 to 2.5 M (Figure 7). Additionally, the extraction efficiencies of Ni and Fe were observed to be <10% in 1.0 M TFMS lixiviant, while that of Fe increased to 18.5% when the concentration of TFMS was increased from 1.0 to 2.5 M. The achieved Cu and Zn extraction of >65% was due to the fact that TFMS as organic acid exhibits almost the same strength as perchloric acids in all solvents [29]. As compared to H$_2$SO$_4$ lixiviant, it is interesting to note that 50% Pb was extracted in 1.0 M TFMS, while the efficiency decreased to 42.1% when the concentration increased to 2.5 M TFMS. The extractions of Al and Sn were observed to be 20%. TFMS has a comparative conductivity of 348.0 S cm$^2$ mol$^{-1}$ [30] to that of hydrochloric acid (346.1 S cm$^2$ mol$^{-1}$) and sulfuric acid (444.9 S cm$^2$ mol$^{-1}$), and higher than methanesulfonic acid (299.6 S cm$^2$ mol$^{-1}$) reported for the efficient recovery of metals from solution using electro-winning [30]. The lower extraction efficiency of Cu and Zn was due to the incomplete dissociation of TFMS in aqueous solution, irrespective of the concentration as a result of its sulfonic end-group [31]. Therefore, TFMS can be used for the selective extraction of Cu and Zn, owing to aforementioned advantages and having co-extracted minimal metals as impurities.

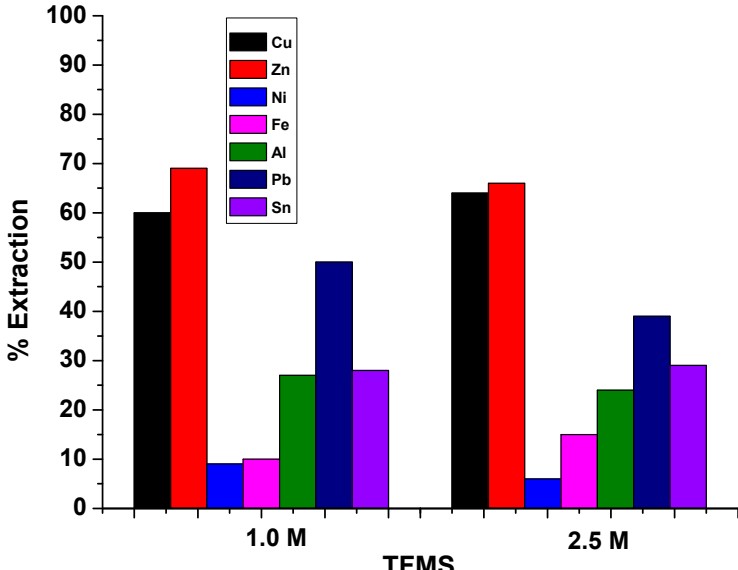

**Figure 7.** Comparative extraction efficiencies of Cu and Zn, and the co-extraction of other metals from WPCBs using TFMS lixiviant, 50 g/L pulp density, 25 °C, 300 rpm, and 12 h.

*3.6. Extraction in Chloride Media*

Figure 8 demonstrates that 33.5% Cu extraction efficiency in 0.5 M NaCl + 0.1 M CuCl$_2$ lixiviant was achieved. When the concentration of CuCl$_2$ in the lixiviant was increased i.e., (0.5 M NaCl + 0.5 M CuCl$_2$), the Cu extraction efficiency decreased drastically while relative increase in the extraction efficiency Zn was achieved. Further increases in the concentration of CuCl$_2$ to 1.0 M and NaCl to 4.5 M (4.5 M NaCl + 1.0 M CuCl$_2$) resulted in further decrease of Cu extraction efficiency (2.0%).The highest Cu selective extraction (33.0%), observed in 0.5 M NaCl + 0.1 M CuCl$_2$ lixiviant was due to chloride ions stabilized Cu (II) ions, thereby increasing Cu solubility. It can be noticed that the minimum concentration of NaCl solution (0.1 M) had an effect on the dissolution of Cu. Carneiro and Leao (2007) also observed a beneficial effect of NaCl on chalcopyrite residue leaching using FeCl$_3$. However, the effect was less pronounced [32].

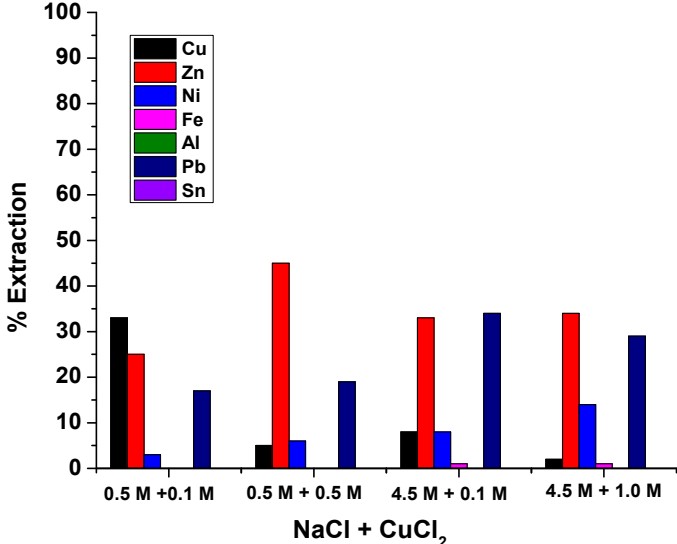

**Figure 8.** Comparative extraction efficiencies of Cu and Zn, and the co-extraction of other metals from WPCBs using NaCl + CuCl$_2$ lixiviant, 50 g/L pulp density, 25 °C, 300 rpm, and 12 h.

On the other hand, a Zn extraction efficiency of 45.0% was achieved with 0.5 M NaCl + 0.5 M CuCl$_2$ lixiviant, while 30% was achieved in other chloride lixiviants. Furthermore, a trace of Fe was observed in chloride media. That is, Fe (III) forms FeCl$^{2+}$ and Fe$^{3+}$ at lower Cl$^-$ concentrations, whereas FeCl$_2$$^+$ is formed at higher chloride concentrations [33–35]. There was no extraction of Fe in lixiviants containing 0.5 M NaCl, while low extraction efficiency was achieved by increasing the concentration to 4.5 M NaCl. Overall, the obtained results show that Fe dissolution was strongly related to the solution pH. At pH < 2, iron is known to remain soluble [36]. This is supported by Equations (4)–(7) below [37]. The extraction of Ni and Pb increased with increased chloride lixiviants.

$$Fe + H_2O = Fe\ (OH)_{ads} + H^+ \tag{4}$$

$$Fe + Cl^- = Fe\ (Cl^-)_{ads} \tag{5}$$

$$Fe\ (OH)ads + Fe\ (Cl^-)\ ads = Fe + FeOH^+\ (Cl^-) + 2e^- \tag{6}$$

$$FeOH^+ + H^+ = Fe^{2+}aq + H_2O \tag{7}$$

### 3.7. Extraction in Hydroxide Media

The selective extraction of Cu and Zn, and the co-extraction of other metals from powdered WPCBs sample was carried out at ambient temperature using different concentrations of NaOH lixiviants (Figure 9).

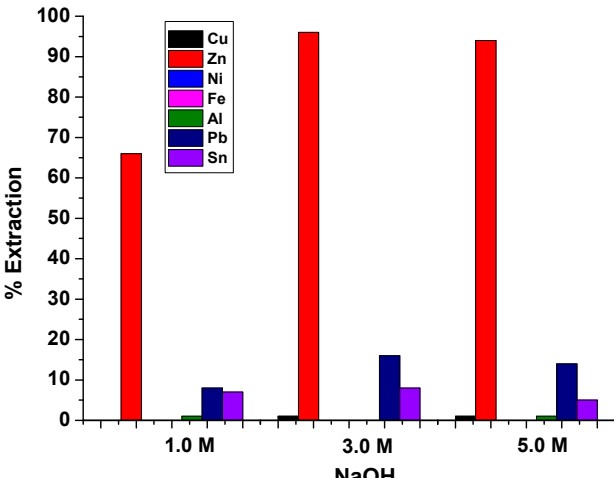

**Figure 9.** Comparative extraction efficiencies of Cu and Zn, and the co-extraction of other metals from WPCBs using NaOH lixiviant, 50 g/L pulp density, 25 °C, 300 rpm, and 12 h.

The highest Zn extraction efficiency was achieved in NaOH lixiviants, and the extraction efficiency increased with increasing concentration from 1.0 to 3.0 M. A further increase in the concentration had a negligible effect on the extraction efficiency. Zinc was selectively dissolved over Cu with traces of Pb and Sn as impurities. However, Zn is amphoteric, it can act as an acid and react with a strong base such as NaOH to form sodium zincate (Equation (8). The same factor applied to Sn, although its dissolution efficiency was ≤10%.

$$ZnO + H_2O + NaOH \rightarrow NaZn(OH)_3 + H_2 \tag{8}$$

According to the XRD results of the powdered WPCBs sample, ZnO and ZnSiO$_4$ were the predominant phases (Figure 1). The highest extraction of Zn as NaZn(OH)$_3$ in NaOH lixiviants was supported by the report by the literature that ZnO solid exists in the equilibrium state in the low-concentration alkali regions and the solubility of zinc oxide is almost invariable with temperature.

When alkali concentration was increased, there was suddenly shift in the equilibrium as reactant ZnO reacted with hydroxide to form product $NaZn(OH)_3$, which indicated higher dissolution of Zn [37].

## 4. Selectivity Studies

This experiment was purposely conducted to obtain suitable lixiviants that would exhibit high selectivity for Cu and Zn while retaining Fe and Ni with other metals in the residues for further hydrometallurgical recovery. Since the extraction of other metals was observed to increase with increases in the concentration, extraction efficiencies of metals with dilute lixiviants were used to calculate the selectivity ratio, as tabulated in Figure 10. Additionally, a selectivity ratio of 5 was set as the minimum standard ratio for the lixiviants to be judged as effective for the selective extraction of Cu and Zn towards Fe and Ni. It is shown in Figure 10 that the selected dilute HCl lixiviants had selectivity ratios below 5. Therefore, HCl lixiviant exhibit poor selectivity, as the leach solution obtained contains higher other metals as impurities which might require a tedious subsequent purification process to recover. It is interesting to note that Cu/Fe and Zn/Fe ratios using both 0.5 and 1.5 M $H_2SO_4$ were ≥5, while Cu/Ni and Zn/Ni were ≥10. An excellent selectivity ratio (≥10) was achieved by 1.0 M $HNO_3$ and 0.5 M + 0.1 M NaCl + $CuCl_2$ lixiviants. Similarly, 2.5 M $HNO_3$ offered moderate selectivity (≥5) for both Cu and Zn against Fe and Ni. In TFMS lixiviants, selectivity ratios (≥5) for Cu/Fe and Zn/Fe were achieved using 1.0 M.

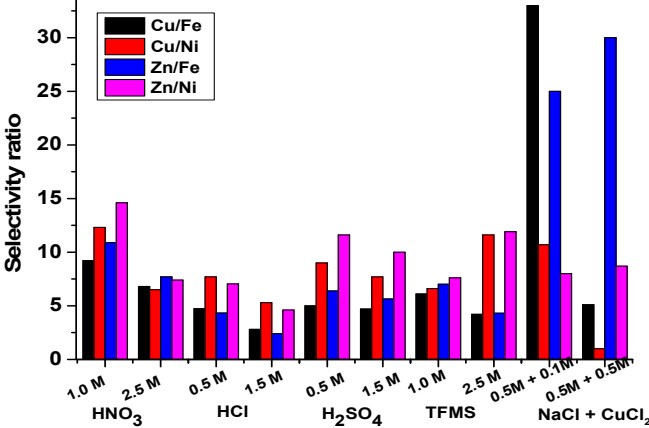

**Figure 10.** Plot of selectivity ratios for the selected dilute lixiviants.

## 5. Conclusions

This study demonstrated that Cu and Zn could be selectively extracted from WPCBs using diluted lixiviants at longer time. The extraction capacities of each lixiviant were investigated at different concentrations with constant time, mixing intensity, and solid–liquid ratio. The following can be concluded:

- Cu and Zn were extracted with other metals in diluted HCl lixiviant, which resulted in a low selectivity ratio and made HCl lixiviants perform poorly for selective extraction.
- Diluted $H_2SO_4$, $HNO_3$, and TFMS lixiviants performed excellently, with the highest selectivity of Cu and Zn toward Fe.
- The selective extraction of Zn without the extraction of other metals was achieved in NaOH lixiviants because Zn acted as an amphoteric oxide. The high selectivity of Zn toward Fe was achieved in 0.5 M NaCl + 0.5M $CuCl_2$ lixiviant.
- Selectivity of Cu and Zn toward Fe in 0.5 M NaCl + 0.1 M $CuCl_2$ lixiviant was obtained. As the concentration of $CuCl_2$ increased to 0.5 M, the Cu extraction efficiency decreased drastically.

Therefore, diluted $H_2SO_4$, $HNO_3$, TFMS, and 0.5 M NaCl + 0.1 M $CuCl_2$ produced pregnant leach solutions rich in Cu and Zn with traces of other metals. This work also provides a foundation which

substantiates further research into the performance of TFMS for the extraction of metals' value, since its application has not been reported previously.

**Author Contributions:** Conceptualization, A.E.A., O.A.A., O.J.A.; Methodology, A.E.A.; Formal Analysis, M.K.G., A.E.A.; Investigation, A.E.A.; Original draft A.E.A., F.E.O.; Review and Editing, M.K.G.; Supervision, S.B. and M.K.G.; Project administration, S.B.

**Funding:** This research received no external funding.

**Acknowledgments:** The authors gratefully acknowledge the Council of Scientific and Industrial Research (CSIR) India, as well as The World Academy of Science (TWAS), Italy, for the award of Doctoral fellowship to Ajiboye Emmanuel Ayorinde.

**Conflicts of Interest:** The authors declare no conflict of interest.

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
