# Peer review of "Extraction of Copper and Zinc from Waste Printed Circuit Boards"

_recycling, doi:10.3390/recycling4030036_

Round 1
Reviewer 1 Report
The paper deals with extraction of copper and zinc and other metals from e-waste materials using commonly available leaching agents. The paper presents some information of interest to the readers but it requires extensive revision.
Among the major concerns related to the published results one may list the following issues:
1. The word “selective” used in the tile of the ms. seems to be rather inadequate, since considerable amounts of other metals are leached by concentrated aqueous solutions of HCl, H2SO4, and HNO3.
2. Extraction of metals from powdered WPCBs was performed at 25oC. How did the authors avoid the significant increase in the system temperature while treating the samples with strong acids or bases? How was the temperature kept at the constant level?
3. Extraction efficiency defined by eq. 3 is not clear. All variables have to be clearly defined and appropriate units have to assigned to them.
4. The chemistry exploited for the leaching process and presented in sections 3.1, 3.2, and 3.3 is well known and no new insights are offered. The technique is merely applied in a novel context. These sections should be significantly shortened.
Some specific comments:
1. Section 3.2.: Sulfuric (VI) acid is characterized by two Ka values.
2. Section 3.3.: The sentence “It would be as result of insufficient hydration needed for complete dissociation and enhances migrations of ions” is totally unclear. Migration refers to the ionic transport in electric field.
3. Section 3.4.: What do the authors mean by “The lower extraction efficiency of Cu and Zn was due to the report of incomplete dissociation of TFMS in aqueous solution irrespective of the concentration as a result of its sulfonic end group, until the monohydrate molten salt is form”. What kind of molten salt is formed?
Author Response
Dear Reviewer,
We thank you for your valuable comments and suggestion in order to make this article better for publishing. We have thoroughly reviewed the text and your suggestions have been included accordingly. Our responses to your comments have been appended in the attached file.

Reviewer 2 Report
Hi,
Interesting study. A few comments:
Figure 2 is an odd presentation. It just shows possible contributors towards peaks. Why was the pattern not solved for relative amounts of each compound? Some of the listed compounds might not even be present. I think there are nearly 20 compounds listed. This XRD pattern needs to be handled properly.
Then in 3.1 it says that there were 8 main phases in the XRD. 2 of these are Ni phases. How can this be, when Ni is a minuscule 0.36 % of the material (Fig. 3 ZAF)? Impossible.
I checked results in Fig. 10 and found some problems. HCL 1.5 M shows about 5 for Cu/Fe selectivity ratio. But if I go to Fig. 4, I find about 70% extraction for Cu and about 25% extraction for Fe. That's a ratio more like 3.
These quick checks, unless I am misunderstanding the data, give me little confidence in the presentation. Perhaps it would be best for the authors to recheck their Figures.
Author Response

(The authors gave the same response as above.)

Reviewer 3 Report
Authors present a work related with the leaching experiments of WPCB´s using different leaching reagents, and indeed this is an interesting work due to the target of minimize this kind of waste that nowadays is an incresing environemtal problem. However, they need to improve some aspects to the paper could be published
Lines 44 to 46 talk about a process that authors are carrying out in this paper? If so, I think that this could be described at the end of the introduction.
During preparation of materials to be leached, what happened with gases produced during roasting? Due there is organic materials, the burning of it could be contaminant without a strict control.
Why authors did only an analysis of particle size by laser diffraction. I believe that a sieve analysis and the treatment of the results by some models could describe well the efficiency of grinding because during leaching experiments the particle size could be determinant to describe the nature of the reaction and some possible kinetics parameters. Which will help to optimize the process
Figure 2 shows the XRD spectra obtained and there can be seen a lot of phases described by author, but there are too much phases only identified with a single peak, which are not the most adequate way to do it well. To identify a mineral phase at least we need to have 3 to 5 family peaks that represent the most representative planes of that specie. If Authors did a Rietveld analysis, there they had to find the real phases present in the material. I cannot see the Rietveld results.
Figure 3 show the SEM - EDX analysis and an image of the particles where this analysis was executed. However, this analysis is semi quantitavive and punctual, so author have to show where these punctual analysis were done pointing in the image the areas of analysis and they have to do an average of the composition found. The mage shows different morphologies and shapes, and they could have different compositions.
Table 1 shows the chemical composition obtained by ICP-OES and AAS, but is this an average composition of both techniques? if so, results could be different because these techniques have different principles. So I think authors need to show the results obtained in both techniques separatly and then they can show the average.
How the concentrations of leaching reagents were selected?
The results show according effciency of leaching reagents are good, but what will happen with the final leaching liqours? They wil be discarded or could be reused, if yes the last, how could be possible?
Author Response

(The authors gave the same response as above.)

Round 2
Reviewer 1 Report
The authors addressed the points raised in the previous review.
Author Response
We thank the reviewer for your comment. The Fig.2. has now been modified.We hope our modification will be find suitable.

Reviewer 2 Report
The problem with the XRD patterns has not been solved and must be addressed before the paper is acceptable. Apparently none of the research team understands that the XRD pattern consists of peaks at specific diffraction angles and specific intensities associated with each of those peaks. For example, the two supposed PbO peaks at 68 and 90 if real would be accompanied by a number of much larger peaks at lower angles. Where are they? Further, the sample supposedly contains around 1% Pb. Conventional XRD essentially will detect a compound containing a component with so little mass.
The alleged Ni(TiO3) is incredibly small and coincides with a minor CuO peak (the 110 peak). We know CuO is in the sample - its major peaks are present. I could not find any of the other required Ni... peaks - they may be obscured by background. Same problem as Pb - 2% Ni and less than 0.2% Ti. Even if all the Ti is in Ni(TiO3) compound no way that shows up on XRD.
When the fundamental composition of the material being studied is in doubt, the entire study becomes questionable.
Author Response
We thank the reviewer for their time and comments. We have modified the Fig.2 in contention. We hope it meet up in line with suggestion.
